# COVID-19 Pandemic-Related Sleep and Mental Health Disparities among Students at a Hispanic and Minority-Serving Institution

**DOI:** 10.3390/ijerph19116900

**Published:** 2022-06-04

**Authors:** Monideepa B. Becerra, Rushil J. Gumasana, Jasmine A. Mitchell, Jeffrey Bao Truong, Benjamin J. Becerra

**Affiliations:** Center for Health Equity, California State University-San Bernardino, San Bernardino, CA 92407, USA; rushil.gumasana@csusb.edu (R.J.G.); 007438243@coyote.csusb.edu (J.A.M.); jeffrey.truong@csusb.edu (J.B.T.); benjamin.becerra@csusb.edu (B.J.B.)

**Keywords:** sleep health, mental health, COVID, pandemic, college students, young adults, minority serving institution, Hispanic serving institution

## Abstract

Background: The COVID-19 pandemic has impacted nearly all sectors of our population, including college students, who continue to share disproportionate rate of disparities. In this study, we aimed to identity key sleep health characteristics, including markers for obstructive sleep apnea, as well as its relation to mental health, physical health, and academic performance. Methods: A cross-sectional study design with online survey dissemination was used. Descriptive, bivariate, and multivariable binary logistic regression analyses were conducted among a predominantly minority population. Results: Results show that nearly 78% of the population reported that the pandemic impacted their mental or physical health, while over 83% reported daytime tiredness/fatigue/sleepiness, and another 61% reported sleeping less than seven hours during weekdays. Among other associations, pandemic-related poor sleep health, including sleeping less than seven hours, was associated with daytime tiredness/fatigue/sleepiness, psychological distress, as well as low mental health and physical health. A severe marker for obstructive sleep apnea, having stopped breathing during sleep, was also associated with psychological distress during the pandemic. Conclusions: Sleep health interventions are critical for optimizing college student health and well-being, including improving mental health outcomes.

## 1. Introduction

The Centers for Disease Control and Prevention (CDC) defines a pandemic as any “event in which a disease spreads across several countries and affects a large number of people” [1]. On December 2019, the first few cases of a novel pneumonia of unknown etiology were first reported in Wuhan, Hubei Providence, China, and by 11 February 2020, the World Health Organization (WHO) announced this as coronavirus disease 2019 (COVID-19) [2,3]. As of March 2022, over 460 million confirmed cases of COVID-19 have been reported globally, in addition to over six million confirmed deaths [3]. In the United States alone, nearly 80 million cases and over 960,000 deaths have been attributed to COVID-19 [4]. Further, while vaccination remains the forefront of preventive measure, young adults have one of the lowest vaccination rates among adult populations in the nation [4].

Furthermore, during the 2020 school year, at the height of the COVID-19 pandemic, most college students had to transition from in-person to virtual class structures. Studies among a college population in the UK have noted heightened prevalence of depression and anxiety, as well as low resilience [5]. Likewise, a review of the existing literature further noted pandemic-related anxiety, depression, as well as altered sleeping habits among college students [6]. Similarly, altered sleep habits, especially duration, have been noted; although the results are not consistent across studies [6,7]. Among other populations, sleep disturbances have also been noted. For example, in a mixed-methods analysis, Krupa et al. noted that patients in isolation due to suspected COVID-19 infection reported feelings of poor sleep health and anxiety [8]. While such studies during the early phase of the pandemic highlighted putative health disparities among college studies, few have addressed sleep health beyond that of duration, and even more so, its role in psychological distress, which is a marker for long-term mental illness. In this study, we aimed to not only assess duration of sleep and the college student’s perception of how the pandemic impacted their sleep hours and quality, but also assess key markers for obstructive sleep apnea development, which has long-term implications for poor health. In particular, we assessed snoring and having stopped breathing during sleep as sleep apnea markers [9], which has been previously used as well [10]. A secondary objective of the study was to assess how such markers are related to physical health among the target population.

## 2. Methods

### 2.1. Data Collection

Data were collected from a variety of undergraduate and graduate courses at a federally designated Hispanic- and minority-serving four year public institution, where a majority of students are Hispanic/Latino (66%), females (63%), undergraduate (89%) and reside in the immediate service area (87%) [11]. Instructors for such courses were asked to share the virtual survey instrument with their students and provide five points extra credit as incentive. No courses where students may have overlapped enrollment were included to ensure only unique participants as survey respondents. Further, inclusion of general education courses also provided a diversity of majors and representation across the target population. Inclusion criteria were all enrolled students aged 18 years or older, students in classes where instructors were willing to provide extra credit for participation, students who consented to participate, and currently enrolled students (full-time and part-time). Our response rate was approximately 80%, although some courses did not track the total. Students were given a consent form, outlining the purpose, contact information, as well as institutional required components. Only those who gave consent to participate were included in the study. All data were collected via an online survey collection tool, and an unlinked page was used to collect students’ names, course numbers, and instructors’ names to provide incentive. All data were downloaded into Excel and transferred to SPSS v28 (IBM Corp.; Armonk, NY, USA) for analysis.

### 2.2. Measures

The primary variables of interest in our study were sleep health and mental health (including psychological distress). We measured sleep using the Berlin sleep questionnaire [12,13], with modifications, as previously published [10], and we added additional measures of how participants felt their sleep quality and hours were impacted due to the on-going COVID-19 pandemic. Responding “yes” to the pandemic lowering the quality of sleep or the pandemic lowering the hours of sleep were coded into the new variable: low sleep health due to the pandemic.

Next, a general mental health status question was designed based on the California Health Interview Survey to allow for comparison across populations. The variable was dichotomized as poor mental health status (reporting average, poor, or very poor) versus good mental health status (reporting excellent or good) to the question “How would you describe your general mental health status?”. We further added two additional questions to assess students’ perception of whether their mental health got worse due to COVID-19-related isolation and whether their mental health got worse due to COVID-19-related diagnosis or symptoms. Responding “average/poor/very poor” or “yes” to any of the above questions, respectively, was coded to create the variable: low mental health status due to the pandemic.

In addition, we also measured psychological distress using the Kessler-6 scale [14], a validated instrument to assess risk of serious mental illness in the general population. Given that our questions on general mental health status are a short-term assessment and Kessler-6 scales each provide different dimensions of mental health assessment, we did not combine the questions into a larger scale, to allow for disaggregation of the short-term versus long-term mental health burden.

Likewise, a general physical health status question was designed based on the California Health Interview Survey to allow for comparison across populations. The variable was dichotomized as poor physical health status (reporting average, poor, or very poor) versus not poor physical health status (reporting excellent or good) to the question “How would you describe your general mental health status?”. We further added two additional questions to assess students’ perception of whether their physical health got worse due to COVID-19-related isolation and whether their physical health got worse due to COVID-19-related diagnosis or symptoms. Responding “average/poor/very poor” or “yes” to any of the above questions was coded to create the variable: low physical health status due to the pandemic.

We also asked demographic characteristics of sex at birth (male, female, intersex/other), age, and race/ethnicity. We did not have any participant reporting sex at birth other than male and female, and thus, the variable was dichotomized in all analyses. Further, due to high prevalence of Hispanic ethnicity in our study, we dichotomized our race/ethnicity variable to that of ethnicity only (Hispanic vs. non-Hispanic). In addition, while we did collect academic grade level (first year, second year, etc.), due to the institution having non-traditional students, where grade level and age do not often align, we chose to include age (years) for analyses.

### 2.3. Data Analyses

To identify the study population characteristics, descriptive statistics were conducted. Next, to assess which key sleep characteristics (hours, tiredness/fatigue/sleepiness, stopped breathing, or snoring) were impacted among participants who noted that the pandemic negatively influenced their sleep health, chi-square tests for association were conducted, followed by multivariable binary logistic regression analyses. Likewise, chi-square tests followed by multivariable binary logistic regression analyses were used to assess how such sleep characteristics influenced mental health and physical health of our participants. All data analyses were conducted in SPSS v28 (IBM Corp.; Armonk, NY, USA), and an alpha of 0.05 was used to denote significance. This study was approved by the Institutional Review Board.

## 3. Results

### 3.1. Study Population Characteristics

Table 1 displays the study population characteristics. We excluded all respondents who only completed less than 50% of the survey. These included students who only completed half the survey and stopped. Given the variables of interest in this study, we spread out throughout the survey to ensure complete data collection; we, thus, excluded the five that did not complete the full survey. As such, a total of 212 participants were included in the study for analyses, with a higher percentage being females, Hispanic/Latino, and 21–23 years of age; thus, this was similar to the institution’s demographics.

When evaluating our sleep health variables of interest, 51.7% of the participants reported low sleep health as a result of the pandemic, 61.3% of participants reported getting less than seven hours sleep, and 83.3% of participants reported daytime tiredness/fatigue/sleepiness. In addition, among markers for sleep apnea risk, 8.6% and 46.4% noted that either they or someone else noticed that they stopped breathing during sleep or were snoring during sleep, respectively. Furthermore, among our mental and physical health variables of interest, 53.1% of the participants had psychological distress, while 77.8% reported that either mental or physical health were negatively impacted by the pandemic.

### 3.2. Sleep Health

Next, we aimed to assess what key sleep characteristics were negatively impacted among those who reported that the pandemic lowered their sleep health. As shown in Table 2, participants with low sleep health (versus those without) had a significantly higher prevalence of sleeping less than seven hours a night (81.5% vs. 41.6%, *p* < 0.05) and daytime tiredness/fatigue/sleepiness (94.4% vs. 71.3%, *p* < 0.05), which were further confirmed in adjusted regression models. Snoring and having stopped breathing, markers for sleep apnea [15,16], did not reach significance with reporting low sleep health due to the pandemic.

### 3.3. Sleep and Mental Health

Table 3 shows significant association between sleep health and mental health variables of interest. Sleeping less than seven hours a night (compared to seven or more hours a night) was significantly associated with reporting psychological distress (62.3% vs. 38.3%, *p* < 0.05) and low mental health status (72.3% vs. 52.4%, *p* < 0.05), with such associations persisting in regression analyses as well. Likewise, reporting daytime tiredness/fatigue/sleepiness (vs. not) was also associated with a significantly higher prevalence of psychological distress (61.5% vs. 14.3%, *p* < 0.05) and low mental health status (71.8% vs. 28.6%, *p* < 0.05), with such associations remaining significant in adjusted regression models.

Further, those who reported that they stopped breathing during sleep (vs. not) also had a higher prevalence of psychological distress (77.8% vs. 51.3%, *p* < 0.05) and low mental health status (88.9% vs. 62.3%, *p* < 0.05), with regression results confirming such associations. Snoring did not yield significance in bivariate or regression models with distress; however, snoring was associated with low mental health status (75.3% vs. 55.4%, *p* < 0.05). Such results persisted in adjusted regression models as well.

In addition, reporting low sleep health during the pandemic (vs. not) was significantly associated with a higher prevalence of psychological distress (68.5% vs. 37.6%, *p* < 0.05) and low mental health (76.9% vs. 51.5%, *p* < 0.05), with adjusted models noting the same associations.

### 3.4. Sleep and Physical Health

Table 4 shows the association between each of the sleep health and physical health variables. Participants who had less than seven hours of sleep (compared to seven or more hours) reported a significantly higher prevalence of low physical health status (73.1% vs. 51.2%, *p* < 0.05). Likewise, increased prevalence of reporting a low physical health status was also found among those who reported daytime tiredness/fatigue/sleepiness (69.5% vs. 40.0%, *p* < 0.05), snoring (75.3% vs. 55.4%, *p* < 0.05), and overall low sleep health due to the pandemic (72.2% vs. 56.4%, *p* < 0.05), compared to their counterparts. Such results remained in regression analyses as well.

We also wanted to assess a putative relationship between reporting a low mental health status and low physical health status due to the pandemic. Results note that among those with psychological distress (vs. no distress), a higher prevalence also had low physical health status (78.8% vs. 50.5%, *p* < 0.05). Likewise, reporting a low mental health status (compared to their counterparts) was also significantly associated with a reported low physical health status (79.6% vs. 37.3%, *p* < 0.05). Results of the multivariable binary logistic regression analyses further confirmed such associations upon adjusting for control variables.

## 4. Discussion

We surveyed a mid-sized federally designated Hispanic- and minority-serving four-year public institution to assess health disparities related to the COVID-19 pandemic. As summarized in Table 5, there are several key findings in our study that warrant a further discussion.

First, our results show that participants who reported that the pandemic worsened their sleep health (quality and/or hours, also had a significantly higher prevalence of reporting sleeping less than seven hours a night and daytime tiredness/fatigue/sleepiness. While similar studies comparing sleep health to such characteristics among our target population are limited, global results demonstrate a similar trend. For example, in a systematic review and meta-analysis analyzing sleep problems during the early parts of the pandemic, Jahrami et al. [17] noted a nearly 36% global pooled prevalence of reporting sleep problems, with the highest rate found among those with COVID-19. Likewise, an assessment across 59 countries during the early phases of pandemic-related lockdown found that a majority of the sample went to sleep later and woke up later, and a third also reported sleep disturbances [18]. Together with the literature, our results not only demonstrated the continued burden of obtaining less than the recommended hours of sleep among college students, but also highlight that more than half the population noted their sleep quality getting worse due to the pandemic, as well as daytime fatigue/tiredness/sleepiness, further proving the insights into the consequences of poor sleep health. In addition, the high prevalence of daytime fatigue/tiredness/sleepiness are further indicative of a need for campus-based initiatives to promote rest among the target population, especially as such daytime sleep deprivation symptoms are associated with cognitive impairment [19].

Next, we also found that overall low sleep health during the pandemic, sleeping less than seven hours a night, daytime tiredness/fatigue/sleepiness, and having stopped breathing were all associated with the presence of psychological distress and low mental health status during the pandemic. While snoring was not associated with psychological distress, it was, however, associated with reporting low mental health status during the pandemic. Gusman et al. [20] found that while switching to distance learning was associated with improved sleep quality early on, the presence and perception of stress was related to lower sleep quality. Furthermore, poor sleep quality and anxiety were also found across additional studies assessing such outcomes among college students in China [21] and medical students in Greece [22]. Such results, in combination with our study, highlight the substantial relation between sleep and mental health. However, our study expands the body of literature by evaluating key sleep characteristics associated with overall poor mental health as well as psychological distress, which are distinct dimensions of assessment [23]. We found all sleep characteristics, except for snoring, to be associated with both overall low mental health and psychological distress. This is likely explained as snoring is a less severe marker for sleep deprivation and is reported to be prevalent at some point in nearly all adults [16]. Thus, longitudinal studies are needed to assess the duration of snoring while sleeping with that of distress.

Further, with the exception of those who reported breathing having stopped during sleep, all other sleep and mental health variables were also associated with reporting low physical health status and a negative impact on academic performance, a pattern noted in the literature as well. For example, among college students, a study conducted on college campuses in mainland China also reported that poor physical health status was associated with more sleep difficulty [24]. Likewise, in another study among Chinese students, sleep duration was shown to be associated with physical and psychological well-being [25]. On the other hand, having stopped breathing during sleep is a severe symptom of obstructive sleep apnea [26] and thus, as a rare case in our population, is less likely to yield statistical significance. In a study among Singapore college students, Armand et al. [27] also found a non-linear relation between mental health, academic performance, and overall sleep quality, further showing the need for more long-term assessment. Nevertheless, future case-control retrospective studies, which are appropriate for rare cases, may be of value to assess the relationship between such severe obstructive sleep apnea symptoms and markers for physical well-being.

Cumulatively, our results not only illustrate the importance of addressing the duration of sleep to improve mental health and physical health among students, but also highlight the importance of comprehensive screening of various dimensions of sleep health (especially daytime fatigue/tiredness/sleepiness and markers for obstructive sleep apnea) and, thus, the need for targeted interventions, such as sleep pods [28] on campuses to ensure daytime wakefulness; sleep health literacy programs to improve overall quality [29,30], including cognitive-behavioral self-help [31]; and frequent screening for early preventive measures. Previous studies, prior to the pandemic, have shown that sleep quality is critical to academic performance among college students and younger groups [10,32,33], and thus, the worsened sleep and associated mental health status reported as a result of the pandemic further demonstrates the heightened burden among such a disparity group.

The results of our study should be interpreted in the context of its limitations. The cross-sectional nature of the data collection limits our ability to assess causal relation. Furthermore, survey designs are susceptible to self-report and recall biases. The results of our study are unique to our population of a predominantly low-income geographic area, where the majority of students are first generation college students, and thus, they likely experience the intersectionality of various stressors. Therefore, the results may not be comparable to campuses that do not demographically match our population. Finally, we did not assess the role of other pandemic-related factors, such as job loss and loss of family members, etc., on our variables of interest, and thus, the true burden on health disparities in the population may be underreported.

Notwithstanding such limitations, there are several strengths to our study. The low–moderate incentive of extra-credit and the inclusion of a variety of majors as well as all those aged 18 years or older limits selection bias. Further, the anonymous nature of the survey reduces the negative impact of social desirability bias, which are common with surveys assessing sensitive topics, such as mental health. Our study questionnaire is primarily also based on previously published and validated instruments and, thus, allows for comparisons across other studies. In addition, our survey questions directly ask about the impact of the pandemic on sleep health, mental health, and physical health, and they do not solely rely on reporting prevalence at the time of pandemic, and instead, they can demonstrate the putative impact of the pandemic on such disparities.

## 5. Conclusions

The results of our study add to the existing body of literature by not only focusing on a predominantly minority student population, but also by addressing the key types of sleep characteristics that were impacted even after a year of the COVID-19 pandemic onset. Further, our assessment goes beyond duration of sleep, as is common in the current literature. In particular, we highlight that in addition to lowered hours of sleep, daytime tiredness, fatigue, and sleepiness were also prevalent among the population. Such additional markers are critical to assess to ensure optimal academic outcomes for students, whose daytime alertness is imperative to positive health and well-being. Although in a smaller proportion, the negative burden of having stopped breathing noted in our study also highlights the need for routine screening among college campuses to implement early prevention strategies for those at risk of developing obstructive sleep apnea. Thus, college campuses aiming to optimize student well-being, including mental health, need to integrate interventions that address sleep health as part of routine assessment.

## Figures and Tables

**Table 1 ijerph-19-06900-t001:** Study population characteristics.

Sex at Birth	
Female	73.1%
Male	26.9%
**Ethnicity**	
Hispanic/Latino	64.9%
Non-Hispanic/Latino	35.1%
**Age (years)**	
18–20	26.9%
21–23	31.7%
24–26	10.6%
27 or more years	30.8%
**Sleep health**	
Less than 7 h of sleep	61.3%
Daytime tiredness/fatigue/sleepiness	83.3%
Stopped breathing during sleep	8.6%
Snoring during sleep	46.4%
Low sleep health	51.7%
**Mental and physical health**	
Psychological distress	53.1%
Low mental health	64.6%
Low physical health	64.6%
Low mental or physical health	77.8%
Low mental and physical health	51.4%

**Table 2 ijerph-19-06900-t002:** Bivariate and adjusted associations between pandemic-related reduction in quality and hours of sleep with sleep health characteristics.

	Slept Less than 7 h	Daytime Tiredness/Fatigue/Sleepiness	Stopped Breathing	Snoring
**Bivariate analyses, %**				
Low sleep health	***	***		
Yes	81.5%	94.4%	11.1%	50.0%
No	41.6%	71.3%	5.9%	42.6%
**Logistic regression analyses, OR (95% CI)**				
Low sleep health	***	***		
Yes	7.39 (3.75, 14.58)	6.11 (2.35, 15.89)	2.40 (0.80, 7.17)	1.41 (0.79, 2.52)
No	Reference	Reference	Reference	Reference

Each logistic regression model was adjusted for age, ethnicity, and sex. *** *p* < 0.001.

**Table 3 ijerph-19-06900-t003:** Bivariate and adjusted associations between sleep health characteristics and mental health outcomes.

	Any Distress (Kessler 8 or More)		Low Mental Health Status	
	Bivariate analyses, %	Regression analyses, OR (95% CI)	Bivariate analyses, %	Regression analyses, OR (95% CI)
**Less than 7 h of sleep**	***	**	**	**
Yes	62.31%	2.72 (1.49, 4.98)	72.3%	2.46 (1.34, 4.52)
No	38.27%	Reference	52.4%	Reference
**Daytime tiredness/fatigue/sleepiness**	***	***	***	***
Yes	61.49%	8.96 (3.24, 24.77)	71.8%	6.37 (2.75, 14.75)
No	14.29%	Reference	28.6%	Reference
**Stop breathing**	*	*	*	*
Yes	77.78%	4.15 (1.24, 13.96)	88.9%	5.44 (1.17, 25.16)
No	51.31%	Reference	62.3%	Reference
**Snoring**			**	**
Yes	60.82%	1.78 (1.00, 3.20)	75.3%	2.46 (1.133, 4.56)
No	47.32%	Reference	55.4%	Reference
**Low sleep health**	***	**	***	***
Yes	68.5%	3.33 (1.85, 5.99)	76.9%	3.03 (1.65, 5.58)
No	37.6%	Reference	51.5%	Reference

Each logistic regression model was adjusted for age, ethnicity, and sex. * *p* < 0.05, ** *p* < 0.01, *** *p* < 0.001.

**Table 4 ijerph-19-06900-t004:** Bivariate and adjusted associations between sleep health characteristics and physical health outcomes.

	Low Physical Health Status	
	Bivariate analyses, %	Regression analyses OR (95% CI)
**Less than 7 h of sleep**	**	**
Yes	73.1%	2.52 (1.36, 4.65)
No	51.2%	Reference
**Daytime tiredness/fatigue/sleepiness**	***	**
Yes	69.5%	3.83 (1.69, 8.67)
No	40.0%	Reference
**Stop breathing**		
Yes	83.3%	3.05 (0.82, 11.36)
No	62.8%	Reference
**Snoring**	**	*
Yes	75.3%	2.20 (1.89, 4.09)
No	55.4%	Reference
**Low sleep health**	*	*
Yes	72.2%	2.19 (1.19, 4.04)
No	56.4%	Reference
**Any distress**	***	***
Yes	78.8%	6.47 (3.35, 12.52)
No	50.5%	Reference
**Low mental health**	***	***
Yes	79.6%	6.47 (3.35, 12.52)
No	37.3%	Reference

Each logistic regression model was adjusted for age, ethnicity, and sex. * *p* < 0.05, ** *p* < 0.01, *** *p* < 0.001.

**Table 5 ijerph-19-06900-t005:** Cumulative results on the association between sleep health, mental health, physical health, and academic performance.

	Distress	Low Mental Health	Low Physical Health
Low sleep health	+	+	+
Less than 7 h per night	+	+	+
Daytime tiredness/fatigue/sleepiness	+	+	+
Stopped breathing	+	×	×
Snoring	+	+	+
Low physical health	+	+	n/a

“+” = positive association, “×” = no association, n/a = not applicable.

## Data Availability

Not available for distribution per Institutional Review Board approval guidelines.

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
