# Peer review of "COVID-19 Pandemic-Related Sleep and Mental Health Disparities among Students at a Hispanic and Minority-Serving Institution"

_ijerph, 2022, doi:10.3390/ijerph19116900_

Round 1
Reviewer 1 Report
This is a survey of federally designated Hispanic and minority serving 4-year public institution to assess health disparities related to COVID 19 pandemic.
51.7% of the participants reported low sleep health as a result of pandemic, 61.3% reported getting less than 7 hours of sleep, 83.3% participants reported daytime tiredness during pandemic.
sleeping less hours associated with psychological distress and low mental health. Daytime tiredness was associated with higher psychological distress and low mental health status. snoring was associated with lower mental health status.
Likewise low physical health noted in patients with low mental health, who sleep less than 7 hrs, people with day time sleeping, people with snoring and any distress.
Many studies proved the same point in the past. This is very good study assessing the college students.
Author Response
Thank you for your feedback. We have read through the document several additional times and addressed the spelling errors. We appreciate your positive comments.
Reviewer 2 Report
I find the study interesting. I present a few remarks below.
- I think that in tables with results, in addition to percentages, N values ​​should be added, indicating the number of responses;
- What were the inclusion and exclusion criteria for the study? Please provide in points;
- I suggest adding a few items from the current literature, which will increase the value of the publication:
Krupa S, Paweł W, Mędrzycka-Dąbrowska W, Lintowska A, Ozga D. Sleep Disturbances in Individuals Quarantined Due to SARS-CoV-2 Pandemic in Poland: A Mixed Methods Design Study. Global Advances in Health and Medicine. January 2021.doi: 10.1177 / 21649561211020707
Armand MA, Biassoni F and Corrias A (2021) Sleep, Well-Being and Academic Performance: A Study in a Singapore Residential College. Front. Psycho. 12: 672238. doi: 10.3389 / fpsyg.2021.672238
Krupa S, Paweł W, Mędrzycka-Dąbrowska W, Lintowska A, Ozga D. Sleep Disturbances in Individuals Quarantined Due to SARS-CoV-2 Pandemic in Poland: A Mixed Methods Design Study. Glob Adv Health Med. 2021 Jun 7; 10: 21649561211020707. doi: 10.1177 / 21649561211020707. PMID: 34164228; PMCID: PMC8188971.
Thank you for the opportunity to review and good luck with publishing.
Author Response
- I think that in tables with results, in addition to percentages, N values ​​should be added, indicating the number of responses
Response: Per the guidelines by which we were approved for this study by our IRB, we can provide cumulative sample size instead of each cell to avoid accidental identification of our student participants.
- What were the inclusion and exclusion criteria for the study? Please provide in points;
Response: We have added the inclusion and exclusion criteria in details in the methods.
- I suggest adding a few items from the current literature, which will increase the value of the publication:
Response: We have included the recommended articles in our background and discussion section.
Reviewer 3 Report
The authors present an original article that aims to identify key characteristics of sleep health, physical activity, and academic performance in students. Below are my comments after reviewing your manuscript.
- The introduction needs more background and theoretical framework. Point out in the introduction the prevention and markers that can be seen in sleep apnea in young adults as this is one of the objectives of your study.
- As these are results whose interpretation is limited, I would describe the pandemic situation of the place where the study was carried out, when were the data collected, were there limitations in mobility or some kind of confinement for the population?
- In the methodology I have some questions. Can we know the total population to which you sent the link for data collection? You state that you have excluded those people who have answered less than 50% of the questions, it is not clear to me if there are items in your questionnaires that some people answer and others do not in order to then carry out the analyses. Does this occur?
- In the tables include the n of the subjects in the same way that I recommend putting the n in brackets after the percentage in the wording of the text of the results. In table 5 I would include a footnote to indicate what the symbols refer to.
- In the discussion I would also point to previous studies, what was the situation like for young people before confinement? were there similar studies?
Author Response
The authors present an original article that aims to identify key characteristics of sleep health, physical activity, and academic performance in students. Below are my comments after reviewing your manuscript.
- The introduction needs more background and theoretical framework. Point out in the introduction the prevention and markers that can be seen in sleep apnea in young adults as this is one of the objectives of your study.
Response: We have updated the introduction section including sleep apnea markers. Given the novelty of covid-19’s impact the current literature does not provide a theoretical framework and thus we have kept the background consistent with other published literature.
- As these are results whose interpretation is limited, I would describe the pandemic situation of the place where the study was carried out, when were the data collected, were there limitations in mobility or some kind of confinement for the population?
Response: Thank you for this valuable insight. We have updated the methods section to discuss the population and pandemic situation in the area.
- In the methodology I have some questions. Can we know the total population to which you sent the link for data collection? You state that you have excluded those people who have answered less than 50% of the questions, it is not clear to me if there are items in your questionnaires that some people answer and others do not in order to then carry out the analyses. Does this occur?
Response: We have clarified this in the methods section and yes due to five students only complete half the survey we could not determine their demographics, which was at the end of the survey.
- In the tables include the n of the subjects in the same way that I recommend putting the n in brackets after the percentage in the wording of the text of the results. In table 5 I would include a footnote to indicate what the symbols refer to.
Response: We have updated the tables accordingly however per the guidelines by which we were approved for this study by our IRB, we can provide cumulative sample size instead of each cell to avoid accidental identification of our student participants.
- In the discussion I would also point to previous studies, what was the situation like for young people before confinement? were there similar studies?
Response: We have included previous studies in the discussion.
Round 2
Reviewer 3 Report
The authors have made the suggested changes. At this stage I believe that all possible improvements have been made.